# L-Arginine Inhibited Inflammatory Response and Oxidative Stress Induced by Lipopolysaccharide via Arginase-1 Signaling in IPEC-J2 Cells

**DOI:** 10.3390/ijms20071800

**Published:** 2019-04-11

**Authors:** Yueqin Qiu, Xuefen Yang, Li Wang, Kaiguo Gao, Zongyong Jiang

**Affiliations:** 1State Key Laboratory of Livestock and Poultry Breeding; Key Laboratory of Animal Nutrition and Feed Science in South China, Ministry of Agriculture; Guangdong Public Laboratory of Animal Breeding and Nutrition; Guangdong Key Laboratory of Animal Breeding and Nutrition; Institute of Animal Science, Guangdong Academy of Agricultural Sciences, Guangzhou 510640, China; qiuyueqin87@126.com (Y.Q.); wangli1@gdaas.cn (L.W.); gaokaiguo@gdaas.cn (K.G.); 2College of Animal Science, South China Agricultural University, Guangzhou 510642, China

**Keywords:** L-arginine, lipopolysaccharide, inflammatory effect, oxidative stress, Arginase-1

## Abstract

This study aimed to explore the effect of L-arginine on lipopolysaccharide (LPS)-induced inflammatory response and oxidative stress in IPEC-2 cells. We found that the expression of toll-like receptor 4 (*TLR4*), myeloid differentiation primary response 88 (*MyD88*), cluster of differentiation 14 *(CD14)*, nuclear factor-kappaBp65 (*NF-κBp65*), chemokine-8 (*IL-8*), tumor necrosis factor (*TNF-α)* and chemokine-6 (*IL-6*) mRNA were significantly increased by LPS. Exposure to LPS induced oxidative stress as reactive oxygen species (ROS) and malonaldehyde (MDA) production were increased while glutathione peroxidase (GSH-Px) were decreased in LPS-treated cells compared to those in the control. LPS administration also effectively induced cell growth inhibition through induction of G0/G1 cell cycle arrest. However, compared with the LPS group, cells co-treatment with L-arginine effectively increased cell viability and promoted the cell cycle into the S phase; L-arginine exhibited an anti-inflammatory effect in alleviating inflammation induced by LPS by reducing the abundance of *TLR4*, *MyD88*, *CD14*, *NF-κBp65*, and *IL-8* transcripts. Cells treated with LPS+L-arginine significantly enhanced the content of GSH-Px, while they decreased the production of ROS and MDA compared with the LPS group. Furthermore, L-arginine increased the activity of arginase-1 (Arg-1), while Arg-1 inhibitor abolished the protection of arginine against LPS-induced inflammation and oxidative stress. Taken together, these results suggested that L-arginine exerted its anti-inflammatory and antioxidant effects to protect IPEC-J2 cells from inflammatory response and oxidative stress challenged by LPS at least partly via the Arg-1 signaling pathway.

## 1. Introduction

Deriving from Gram-negative bacteria, lipopolysaccharide (LPS) is well known to activate the innate immunity and induce inflammatory response [1]. Toll-like receptors (TLRs) are major pattern recognition receptors (PRRs) which recognize a variety of pathogen-associated molecular patterns (PAMPs) from bacteria, and they can mediate intracellular signals in response to PAMPs [2]. To date, many researchers have studied TLR1 to TLR10, and TLR4 was the primary receptor for LPS, thereby conferring the recognition of gram-negative bacteria [3,4,5,6,7,8]. LPS-induced TLR4 signaling pathways showed that TLR4, CD14 and Myd88 complexes were the indispensable components to recognize LPS [9,10,11,12]. Additionally, it has demonstrated that LPS treatment not only affected immune cells such as T cells and macrophage, but also had an impact on many types of epithelial cells through TLR4 signaling, which led to local infections or inflammatory processes [13]. Intestinal epithelial cells (IECs) play an active role in the mucosal immune response to pathogenic bacteria, which mediate the non-specific acute inflammatory response [14]. Moreover, IECs express pattern recognition receptors (PRRs) which recognize pathogen-associated molecular patterns [2]. Schierack et al. has reported that the porcine intestinal epithelial IPEC-J2 cell line, a non-transformed intestinal cell line deriving from jejunal epithelia, provided an excellent in vitro model system for studies of infection processes and interactions between pathogenic and non-pathogenic bacteria and intestinal epithelia of swine [15]. Moreover, it has been reported that LPS also increased oxidative damage by generating ROS, which could result in cell growth inhibition and lipid peroxidation [16].

L-arginine is considered as a conditionally essential amino acid, which is mainly catalyzes by nitric oxide synthase (NOS), arginine decarboxylase and arginase (ARG) and arginase for synthesis of protein and several bioactive molecules such as nitric oxide (NO), proline, creatine, and polyamines [17,18,19]. Recently, there is an increasing emphasis on the improvement of the physiological functions of arginine, such as attenuating intestinal inflammation and oxidative stress [20,21]. Previous studies demonstrated that L-arginine administration in animals and humans with intestinal diseases can decrease intestinal injury, reduce oxidative stress and inflammation and restore mucosal immune homeostasis [22,23]. Coburn et al., have reported that L-arginine administration could decreased intestinal inflammation in murine models of dextran sulfate sodium (DSS)-induced colitis, an experimental model of IBD [24]. L-arginine also showed an inhibitory effect on IL-1β-mediated NF-κB-activation in Caco-2 cells [25]. In addition, Zhang et al., demonstrated that L-arginine protected ovine intestinal epithelial cells from LPS-induced apoptosis through attenuating oxidative stress [26]. However, the precise molecular mechanism of L-arginine alleviated inflammatory response as well as decreased oxidative injury induced by LPS still remains unexplored. L-arginine serves as a degradation substrate for several enzymes in the cells, such as Arg-1. Several studies have demonstrated that Arg-1 exhibited an anti-inflammatory effect and played a beneficial role in inflammatory disease [27,28]. The possible role of Arg-1 in mediating a protection of L-arginine from LPS-challenged damage in IPEC-J2 cells remained unknown. The objective of this study was, thus, to investigate whether L-arginine exerts protection in IPEC-J2 cells against inflammation and oxidative stress induced by LPS and to investigate the possible molecular mechanism.

## 2. Results

### 2.1. Effect of LPS Stimulation on TLR4 Expression and p38 in IPEC-J2 Cells

As shown in Figure 1, we found that LPS stimulation (100 ng/mL) significantly increased the abundance of TLR4 mRNA and the protein level of TLR4. Additionally, LPS significantly increased phosphorylation of P38.

### 2.2. Effect of L-Arginine on Cell Survival and Cell Cycle

As shown in Figure 2a, cells variability result showed that cell survival was significantly inhibited by LPS treatment (*p* < 0.001), however, the effect of LPS on the cell activity was blocked, when simultaneously adding L-arginine (250 μM or 500 μM). We further studied the cell cycle of IPEC-J2 cell exposed to LPS or co-incubation (LPS plus L-arginine). IPEC-J2 cells treated with the indicated concentration of LPS initiated an apparent G0/G1-phase cell cycle arrest (from 55 to 64%) (*p* < 0.001) with concomitant losses from S phase (from 32 to 23%) (*p* < 0.05), as compared with the control group (Figure 2b–d). However, combining treatment of IPEC-J2 cells with LPS and L-arginine (500 μM), the percentage of cells in S phase was sharply increased (*p <* 0.05), whereas a dramatic decrease of cells in G0/G1 phase occurred (*p <* 0.001), and no consistent effect was noted in G2 phase (Figure 2b–e).

### 2.3. Effect of L-Arginine on TLR4, MyD88, CD14, and Pro-Inflammatory Cytokines in LPS-Treated IPEC-J2 Cells

As shown in Figure 3, LPS induced a significant increase in the abundance of *TLR4* and its related genes including *MyD88* and *CD14* transcripts as compared with the control group, but these responses were reduced in the presence of 500 μM L-arginine (*p* < 0.01, *p* < 0.01, and *p* < 0.05, respectively). The expression of *NF-κBp65*, *IL-8*, *IL-6*, and *TNF-α* transcripts were also increased (*p* < 0.05) by LPS. While addition of L-arginine (500 μM) significantly inhibited LPS-induced the expression of *NF-κBp65* and *IL-8* mRNA. Comparison with LPS treatment group, addition of L-arginine also inhibited the abundance of *IL-6* and *TNF-α* transcripts, although there was no significant statistics (Figure 4).

### 2.4. The Effect of L-Arginine on ROS Production and Contents of MDA, Total Superoxide Dismutase (T-SOD) and GSH-Px in LPS-Stimulated IPEC-J2 Cells

As shown in Figure 5, there was strongly increased ROS production in LPS-treated IPEC-J2 cells compared to the control cells. We further observed that the L-arginine (500 μM) had the antioxidant ability to prevent LPS-induced ROS production. The antioxidant effect of L-arginine was further evaluated by the contents of MDA, T-SOD and GSH-Px examination (Figure 6). The content of MDA increased by 70.5% and GSH-Px decreased by 30.03% (*p* < 0.01, *p* < 0.05, respectively) in cells treated just with LPS, compared to controls. The increase in MDA induced by LPS was significantly offset by 500 μmol/L L-arginine. In contrast, the LPS-challenged decrease in GSH-Px was reversed, in a concentration of 500 μmol/L L-arginine. There were no significant effects of LPS treatment or co-treatment with L-arginine on the content of T-SOD.

### 2.5. L-Arginine Enhanced the Activity of Arg-1, While Arg-1 Expression Inhibition Abrogated the Protection of L-Arginine against LPS-Mediated Damage

LPS treatment reduced the abundance of the *Arg-1* transcript and Arg-1 protein level, while L-arginine had an opposite effect on Arg-1, compared with LPS-treated group (*p <* 0.01) (Figure 7a–c). Addition of Arg-1 inhibitor, the protective effect of L-arginine against LPS-induced high level of *TNF-ɑ* transcript and its protein was abrogated (Figure 7d–f). Additionally, co-treatment with L-arginine significantly decreased the content of MDA in LPS-treated cells, while this antioxidative effect of L-arginine was abolished after Arg-1 inhibitor administration (Figure 7g).

## 3. Discussion

Since all the genes previously reported to be expressed in epithelial cells were detected in IPEC-J2 cells, suggesting that IPEC-J2 cell line has retained most of their original epithelial nature, thus this cell line was considered as valuable models for studies of host-pathogen interaction in pigs [29]. Despite previous studies investigating pathogen responses in IPEC-J2 cells focusing on enterotoxigenic *Escherichia coli* (ETEC) [30,31], *Salmonella*, including *Salmonella enterica* serovar Choleraesuis (SC) or Typhimurium (ST) [32,33,34,35], and purified LPS [33], and focused on the ability of the probiotic to inhibit pathogen-induced inflammatory responses [36], very little work has evaluated effect of L-arginine on these pathogen-induced innate immune response and oxidative stress, despite its wide use in vivo. The present study was well documented TLR4 and related molecules in LPS-treated IPEC-J2 cells. The current data demonstrated that LPS significantly increased the expression of TLR4. Similar to the previous studies, our result suggested that TLR4 was the key receptor of LPS. Contrarily, Burkey et al. described that expression of TLR4 was largely unaffected by LPS (10 ng/mL) after direct apical exposure of the IPEC-J2 cells [32]. Farkas also has been reported that the experiments were repeated under different conditions (10 μg/mL LPS derived from *E. coli*, 1 h and 3 h incubation after treatment, respectively) treatment did not change the relative expression of TLR4 in IPEC-J2 cells [37]. This was not consistent with our results presumably due to the higher LPS concentration and longer exposure time our experiment used. In our work, it was showed that the mRNA expressions of *CD14* and *Myd88* were also enhanced by LPS stimulation, which might suggest that LPS could activate NF-κB and pro-inflammatory cytokine IL-8 through TLR4 and related molecules in cultured IPEC-J2 cells. This observation was consistent with the recent studies, revealing that a significant increase of IL-8 concentration after LPS treatment was observed in the IPEC-J2 monoculture [37,38,39] treated with 10 ng/mL LPS, has been reported [40]. In fact, Arce et al. demonstrated that no activation was observed with the lower concentration of LPS used (10 ng/mL) [41]. As we know, the LPS receptor TLR4 is a central player in signaling pathways of the innate immune response to infection by pathogen [42,43]. Activation of TLR4 by its ligand subsequently was able to induce the expression of inflammatory genes as well as regulated cell growth and/or apoptosis. However, improper modulation or compromised function of TLR4 may contribute to inflammatory diseases. Increasing experimental evidence has demonstrated that chronic infection and inflammatory processes may lead to tumor genesis, and the tumor genesis was mediated in part through recognition of stimuli by TLR4 [43,44,45], therefore, proper expression of TLR4 was very important to the immune system. We found that L-arginine had the ability to modulate TLR4 expression and its related molecules and exerted anti-inflammatory effect in vitro. It might suggest that the L-arginine had the ability to defense chronic infection and inflammatory processes. Our results were consistent with the work by Meng et al. who found that L-arginine had an anti-inflammatory property [25].

An imbalance between antioxidants and pro-oxidants would induce oxidative stress. When severe oxidative stress happens, cell death and irreparable oxidative injury can be induced [46,47,48]. Previous study has provided evidence that LPS induced oxidative stress by generating a large amount of ROS, which may cause cell damage [49]. High levels of pro-inflammatory cytokines, such as TNF-α have been observed as early as 12–24 h after cutaneous injury and had the ability to induce elevated levels of ROS at the wound site [50]. Similar to these results, the result obtained here showed that a higher level of ROS was raised by LPS stimulation in IPEC-J2 cells. In agreement with the work by Yuan [51], we supposed that inflammation signaling activation might contributed to enhancing ROS production by LPS in IPEC-J2 cell. MDA has been used as a biomarker to assess the oxidative stress within cultured cells [52]. The current study demonstrated that MDA level was increased after LPS treatment, suggesting that IPEC-J2 cells underwent the oxidative stress. The antioxidant enzymes including GSH-Px and SOD play an essential role in keeping cellular redox homeostasis [26,53]. In the present study, it was observed that L-arginine was capable of reducing ROS and MDA level, when LPS stimulated IPEC-J2 cells, and activating antioxidant enzyme GSH-Px. These results indicated that L-arginine had an effectively protective effect against cells oxidative damage induced by LPS. Our results in consistent with previous research which reported that L-arginine supplementation decreased the content of MDA and up-regulated expression of the anti-oxidative enzymes GSH-Px for alleviating oxidative injury [26]. Previous study has demonstrated that L-arginine supplementation significantly enhanced the activity of SOD [54], however, oxidative stress generated here by LPS in IPEC-J2 cells did not deplete T-SOD, and increasing of L-arginine concentrations had no effect on T-SOD activity. Thus, these results revealed that L-arginine can suppress oxidative stress induced by LPS, but antioxidant activity of L-arginine was not executed through increased SOD activity.

Inflammation and oxidative stress usually promoted cell growth inhibition and even apoptosis [46,55]. In this study, cell cycle distribution data showed that LPS treatment did lead to G1 arrest and prevented the progression of cells into the S phase of the cell cycle. Cells variability assay further observed that the cell survival was inhibited by LPS stimulation. As we know, controlling progression of cells into and through the S phase of the cell cycle was critical for regulating DNA synthesis and cell proliferation. Addition of L-arginine to the LPS-treated IPEC-J2 cell, we found that progression of cells was regulated from G1 into S phase. These data might suggest that L-arginine had antioxidant ability, and effectively kept cell growth in a normal situation.

Arg-1, as an enzyme to catalyze L-arginine, was expressed at high levels in the liver and at low levels in the intestine [56], and contributed to inflammatory response deactivation [28]. We demonstrated here IPEC-J2 cells could express Arg-1, and Arg-1 was suppressed by LPS, whereas it was activated by L-arginine. Arg-1 inhibition could suppress the anti-inflammatory effect of L-arginine against LPS-induced a high level of TNF-ɑ. This result in current study was in agreement with previous study that Arg-1 suppressed TNF-ɑ production induced by LPS in vascular smooth muscle cells. Furthermore, the antioxidative effect of L-arginine on LPS-induced oxidative damage was counteracted after Arg-1 administration, but the reason remains unknown.

In summary, the current study found that TLR4 could recognize LPS in IPEC-J2 cells and provided conclusive evidence that L-arginine had the ability to reduce LPS-induced inflammatory response and possessed strong antioxidant ability at least partly via Arg-1 signaling. Our study might provide that L-arginine can serve as a potential therapy material for porcine inflammatory disease such as inflammatory bowel disease and diarrhea.

## 4. Materials and Methods

### 4.1. Reagents

Dulbecco’s modified Eagle’s F12 ham medium (DMEM-F12), fetal bovine serum (FBS), and antibiotics were procured from Invitrogen (Gibco, Waltham, MA, USA). Plastic culture plates were manufactured by Corning Inc. (Corning, NY, USA). N-hydroxy-nor-L-a rginine (nor-NOHA) was purchased from APExBIO (APExBIO Inc, USA). phospho-p38 MAP Kinase, p38 MAPK (D13E1), TNF-ɑ (D5G9), Arg-1(#9819) and β-actin (13E5) were purchased from Cell Signaling Technology (Cell Signaling Technology, Boston, MA, USA). Unless indicated, all other chemicals were purchased from Sigma-Aldrich (St. Louis, MO, USA).

### 4.2. Cell Culture

The IPEC-J2 cell line, a non-transformed intestinal cell line originally derived from jejunal epithelia isolated from a neonatal, unsuckled piglet and maintained as a continuous culture [8], was kindly supplied by Dr. Guoyao Wu, Faculty of Nutrition and Department of Animal Science, Texas A and M University, 2471 TAMU, College Station, USA.IPEC-J2 cells were cultured in serial passage in uncoated plastic culture flasks (100 mm^2^) in DMEM-F12 containing 5% FBS, penicillin (50 μg/mL), streptomycin (4 μg/mL) and kept in an incubator at 37℃with 5% CO_2_. Medium was changed every two days. At confluence about 80–90%, cells were trypsinized and seeded in six-well cell culture plates with 3 × 10^5^ cells per well. After an overnight culture, then the cells were starved for 6 h in Arg-free DMEM-F12.

### 4.3. Cell Viability

CCK-8 commercial kit was used to measure cell viability according to the protocol of the manufacturer. 100 μL cell suspension (5 × 10^4^ mL^−1^) was added to the wells of 96-well micro-culture plates. After overnight culture, the cells were starved 6 h in 0.2 mL of Arg-free DMEM-F12, then 110 μL of the medium containing 0, 100, 250 or 500 μM L-arginine and 0 or 100 ng/mL LPS was dispensed and incubations continued for another 24 h. Eight wells containing medium only were used for blanking the reader, another eight wells containing cells and medium were used to determine the control cell survival. 10 μL of the tetrazolium substrate was added to each well; then, plates were further incubated at 37 °C for read after incubating for an additional 1 h. The OD of the wells was determined with a microplate spectrophotometer at 450 nm. The cell survival (CS) was calculated by the equation: CS = (OD treated well/mean OD control wells) × 100%.

### 4.4. Cell Cycle Distribution

IPEC-J2 cells exposed to LPS (0 or 100 ng/mL) and LPS-treated cells were co-incubated with L-arginine (0 or 500 μM). After 24 h incubation, cells were harvested by centrifugation and washed with cold PBS. Cells were stained with PI after fixation with 70% ethanol at 4 ℃ overnight. Then the cell cycle distribution was analyzed by flow cytometric analysis.

### 4.5. TLR4 Expression was Analyzed by Flow Cytometry

IPEC-J2 cells exposed to LPS (0 or 100 ng/mL) for 24 h and then were collected and washed with PBS, stained with mAbs against TLR4. After 40 min incubation, these cells were washed twice with PBS and then fixation with 2% paraformaldehyde, analyzed by flow cytometry.

### 4.6. Real-Time Quantitative PCR Analysis

IPEC-J2 cells exposed to LPS (0 or 100 ng/mL) and LPS-treated cells were co-incubated with L-arginine (0, 100, 250 or 500 μM). After 24 h incubation, cells were harvested and total RNA was isolated with Trizol Reagent (Invitrogen Life Technologies, Waltham, MA, USA). All cDNAs were synthesized from 1μg of total RNA using a reverse transcription kit (Takara, Tokyo, Japan) according to the manufacturer’s recommendations. Then the synthesized cDNA was diluted (1:10) and real-time quantitative PCR amplification was performed with SYBR green I (Takara, Tokyo, Japan) and specific primers for pig mRNA sequences (Table 1). Following PCR, an annealing of primers for 30s and an extension at 72 °C for 30s, and cycle number was 39. The β-actin gene was amplified as an internal control. Relative abundance of target genes in treatment culture cells were calculated with the ΔΔCT method using the average ΔCT values of cells from control wells as the reference expression. The ΔΔCT values were expressed as 2^−ΔΔCT^ to obtain relative abundance values.

### 4.7. Determination of ROS Superoxide Levels

To visualize total intracellular levels of ROS superoxide, immunofluorescence assay analysis was performed. IPEC-J2 cells exposed to LPS (0 or 100 ng/mL) and LPS-treated cells were co-incubated with L-arginine (0, 100, 250, or 500 μM). After 24 h incubation, the culture medium was removed and the cells were washed with serum-free DMEM/F12 and incubated with CM-H2DCFDA (Invitrogen Life Technologies, Waltham, MA, USA) at a final concentration of 10 μM in serum-free DMEM/F12 for 20 min at 37℃. Before immunofluorescence assay, the cells were washed twice with serum-free DMEM/F12.

### 4.8. Evaluating the Role of Arg-1 Signaling

IPEC-J2 cells exposed to LPS (0 or 100 ng/mL) and LPS-treated cells were co-incubated with 500μM L-arginine or incubated together with 500 μM L-arginine and Arg-1 inhibitor (nor-NOHA). After 24 h incubation, the culture medium was removed and the cells were harvested for QPCR and Western blot analysis.

### 4.9. Western Blot Analysis of Protein Levels

Cells were rapidly washed twice with ice-cold PBS and then lysed for 30 min at 4 °C. The supernatant was stored at −80 °C and used for Western blot analysis [57].

### 4.10. Antioxidant/Oxidant Indices Analysis

IPEC-J2 cells exposed to LPS (0 or 100 ng/mL) and LPS-treated cells were co-incubated with L-arginine (0, 100, 250, or 500 μM). After 24 h incubation, the culture medium was removed and the cells were harvested and washed twice with ice-cold PBS, then homogenized by sonication in 1 mL PBS and protein was measured with bicinchoninic acid (BCA). The contents of MDA, total superoxide dismutase (T-SOD) and GSH-Px were measured using assay kits in accordance with the protocol of the manufacturer (Nanjing Jiancheng Institute of Bioengineering and Technology Nanjing, China).

### 4.11. Statistical Analysis

Data analysis was performed using GraphPad Prism Version 5 (GraphPad Software, La Jolla, CA USA) and presented as means ± SEM.A *p* value <0.05 in all cases was considered statistically significant (95% confidence interval).

## Figures and Tables

**Figure 1 ijms-20-01800-f001:**
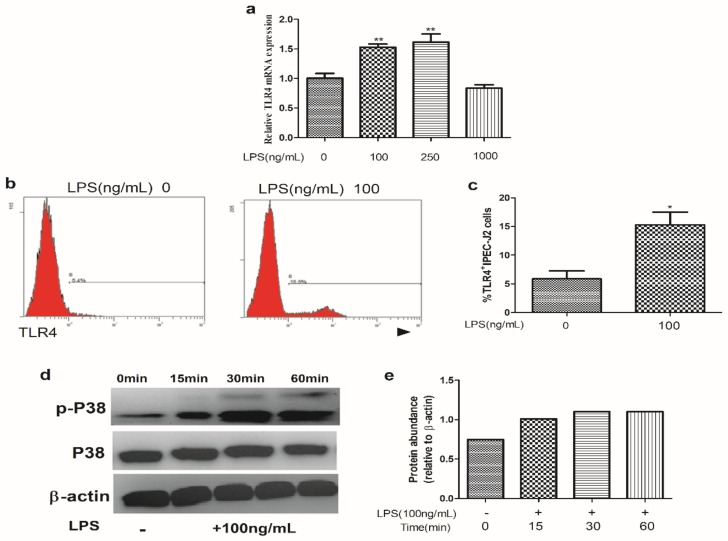
Effect of LPS on TLR4 and P38 expression. (**a**) IPEC-J2 cells were treated with different concentrations of LPS for 24 h, and then were collected for analysis of the relative expression of *TLR4* mRNA. (**b**) Representative flow cytometric histograms demonstrated that the levels of TLR4 in IPEC-J2 cells. (**c**) Summary data showing the expression level of TLR4 in IPEC-J2 cells stimulated with 0 or 100 ng/mL LPS. (**d**) The protein levels of p38 and p-p38 were determined by Western blotting. Equal protein loading was confirmed by analysis of β-actin in the protein extracts. (**e**) The ratios of p-p38 to β-actin in IPEC-J2 cells treated with 100 ng/mL LPS for 0, 15, 30, and 60 min. Data are means ± SEM. * *p* < 0.05, ** *p* < 0.01. TLR4, Toll-like receptor 4.

**Figure 2 ijms-20-01800-f002:**
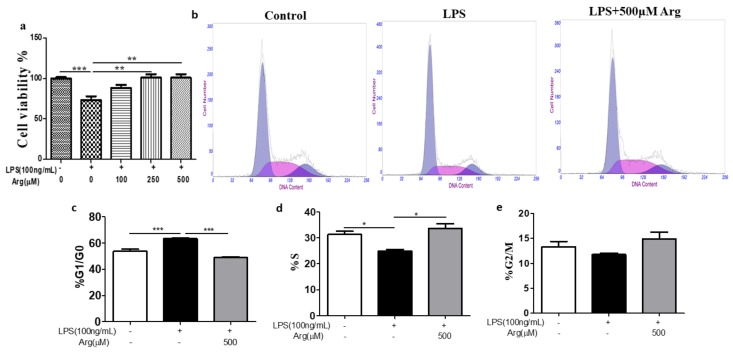
The effect of L-arginine supplementation on LPS-induced cell cycle arrest and cell variability inhibition. (**a**) Cell viability of IPEC-J2 cells (*n* = 8). (**b**) Cells after treatment were collected and stained with Propidium Iodide (PI) solution after fixation by 70% ethanol. Then the DNA content of cells was analyzed by flow cytometry. Representative histograms show regions corresponding to S, G0-G1 and G2-M phases of the cell cycle. Data are from four independent experiments. (**c**)–(**e**) are bar graph data showing that the percentages of S, G0-G1, and G2-M phases of the cell cycle under different treatment, respectively. Data were expressed as mean ± SEM. * *p* < 0.05, *** *p* < 0.001.

**Figure 3 ijms-20-01800-f003:**
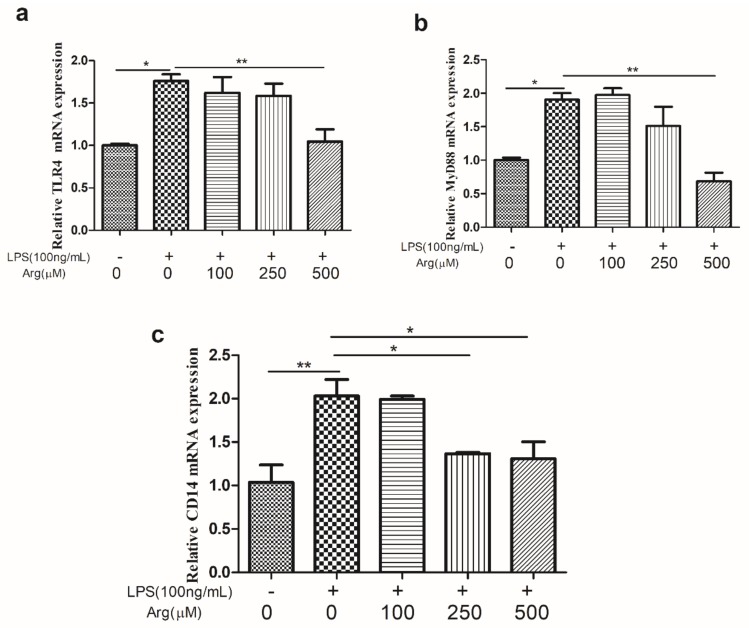
Effect of arginine on *TLR4*, *MyD88* and *CD14* in LPS-treated IPEC-J2 cells. The abundance of (**a**) *TLR4*, (**b**) *CD14*, and (**c**) *Myd88* mRNA in IPEC-J2 cells exposed to LPS (0 or 100 ng/mL) and LPS-treated cells were co-incubated with L-arginine (0, 100, 250, or 50 0μM) for 24 h. Results represent the mean ± SEM from four independent experiments. * *p* < 0.05; ** *p* < 0.01. *TLR4*, Toll-like receptor 4; *CD14*, cluster of differentiation 14; *MyD88*, Myeloid differentiation primary response gene 88.

**Figure 4 ijms-20-01800-f004:**
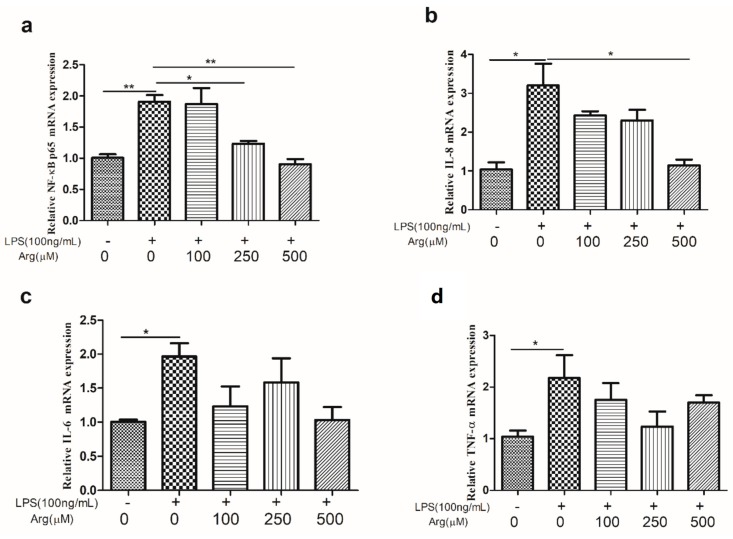
Effect of arginine on pro-inflammatory cytokines in LPS-treated IPEC-J2 cells. The abundance of (**a**) NF-κBp65, (**b**) IL-8, (**c**) IL-6, and (**d**) TNF-ɑ in IPEC-J2 cells exposed to LPS (0 or 100 ng/mL) and LPS-treated cells were co-incubated with arginine (0, 100, 250, or 500 μM) for 24 h. Data were representative of triplicate experiments and expressed as the mean ± SEM.* *p* < 0.05, ** *p* < 0.01. NF-κBp65, nuclear factor-κBp65; IL-8, interleukin-8; IL-6, interleukin-6.

**Figure 5 ijms-20-01800-f005:**
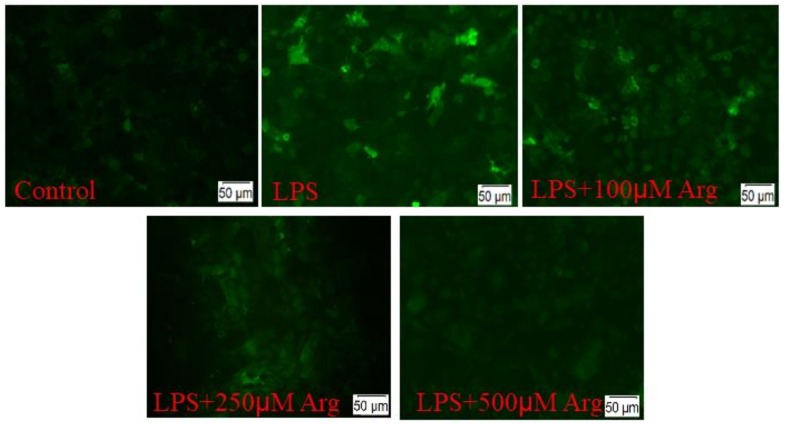
L-Arginine supplementation (Arg) reduced LPS-triggered reactive oxygen species (ROS) in IPEC-J2 cells. Three replications were performed for each experiment.

**Figure 6 ijms-20-01800-f006:**
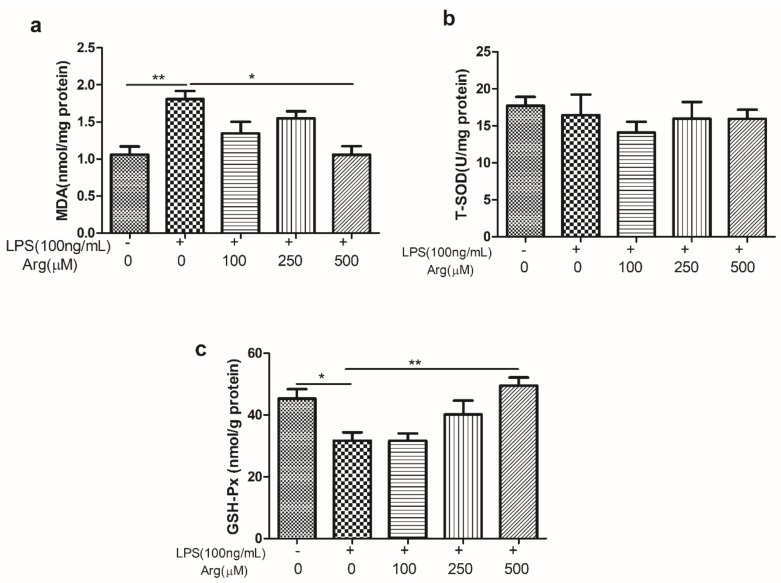
The effect of L-arginine on contents of MDA, T-SOD and GSH-Px in LPS-stimulated IPEC-J2 cell. IPEC-J2 cells exposed to LPS (0 or 100 ng/mL) and LPS-treated cells were co-incubated with L-arginine (0, 100, 250, or 500 μM) for 24 h, then the cells were collected to analyze the contents of (**a**) MDA, (**b**) T-SOD, and (**c**) GSH-Px. Data were representative of triplicate experiments and expressed as the mean ± SEM.* *p* < 0.05, ** *p* < 0.01. MDA, malonaldehyde; GSH-Px, glutathione peroxidase; T-SOD, total superoxide dismutase.

**Figure 7 ijms-20-01800-f007:**
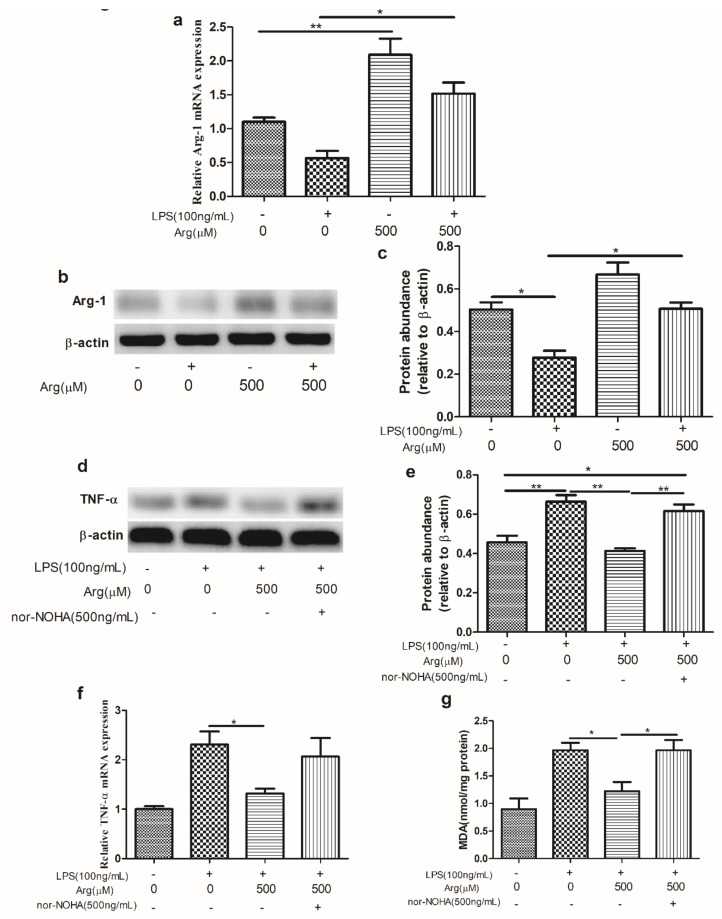
The protection of L-arginine against LPS-induced inflammatory response and oxidative stress via Arg-1 signaling. (**a**) the abundance of *Arg-1* transcript; (**b**) the protein levels of Arg-1 was determined by Western blotting. Equal protein loading was confirmed by analysis of β-actin in the protein extracts; (**c**) the ratio of Arg-1 to β-actin in IPEC-J2 cells treated with LPS, L-arginine, or both. (**d**) The protein levels of TNF-ɑ was determined by Western blotting. Equal protein loading was confirmed by analysis of β-actin in the protein extracts; (**e**) the ratio of Arg-1 to β-actin; (**f**) the abundance of *TNF-ɑ* transcript; (**g**) the content of MDA in IPEC-J2 cells treated with LPS (0 or 100 ng/mL) and LPS-treated cells were co-incubated with L-arginine or L-arginine together with Arg-1 inhibitor. Data were representative of triplicate experiments and expressed as the mean ± SEM.* *p* < 0.05, ***p* < 0.01. Arg-1, arginase-1.

**Table 1 ijms-20-01800-t001:** Primer sequences used in this study.

Gene	Sequences (5′–3′)		Product Size (bp)	GenBank Accession
*TLR4*	Forward Reverse	TGACGCCTTTGTTATCTACTCC GGTCTGGGCAATCTCATACTC	246	NM_001113039
*Myd88*	Forward Reverse	CCCCAGCGATACCCAGTTTGT ATCCGACGGCACCTCTTTTCA	152	NM_001099923
*CD14*	Forward Reverse	GAGTGAGGACAGATAGCGTTTG CTGCGGATGCGTGAAGTT	242	NM_001097445
*IL-8*	Forward Reverse	CTCATTCCTGTGCTGGTCA GCAAGTTGAGGCAAGAAGAC	270	NM_213867
*NF-kappa* *Bp65*	Forward Reverse	ACCCCTTCCAAGTTCCC CCCGAGTTCCGATTCAC	195	NM_001114281
*IL-6*	Forward Reverse	TCAGTCCAGTCGCCTTCT TACCTCCTTGCTGTTTTCAC	227	NM_214399.1
*TNF-ɑ*	Forward Reverse	CACGCTCTTCTGCCTACTGC GTCCCTCGGCTTTGACATT	164	NM_214022.1
*Arginase-1*	Forward Reverse	CTTTCTCCAAGGGTCAGC TCCCCGTAATCTTTCACAT	115	NM_214048.2
*β-actin*	Forward Reverse	CACGCCATCCTGCGTCTGGA AGCACCGTGTTGGCGTAGAG	210	NC_010445

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
