# Peer review of "L-Arginine Inhibited Inflammatory Response and Oxidative Stress Induced by Lipopolysaccharide via Arginase-1 Signaling in IPEC-J2 Cells"

_ijms, 2019, doi:10.3390/ijms20071800_

Reviewer 1 Report

The paper focuses on the analysis of the effect of L-arginine on lipopolysaccharide (LPS)-induced inflammatory response and oxidative stress in IPEC-2 cells. The paper is interesting and acceptable for publication in the journal. However, some minor changes should be addressed by the authors in order to improve the clarity and the understanding.

1- The Introduction should be more referenced. Some references are out of date and should be replaced by some more recent ones. In particular, the authors missed some recent works on L-Arginine functionalization as stabilizer and bioactive molecule.

2- The authors should report the acronyms the first time they appear in the text.

3- Figure 6 is not cited in the text and should be more discussed.

Author Response

April 5, 2019

Dear Professor Wu,

Thank you for your letter and for the reviewers’ comments concerning our manuscript entitled “L-arginine inhibited inflammatory response and oxidative stress induced by lipopolysaccharide via arginase-1 signaling in IPEC-J2 cells” (Manuscript ID: ijms-473911). These comments are valuable and very helpful for revising and improving our paper, as well as the important guiding significance to our researches. We have studied comments carefully and have made the revisions that are highlighted in red in the manuscript, and we hope these revisions can meet with approval. Our responses to the Reviewers’ comments are as flowing:

Reviewers' comments:

Reviewer #1:

Comments and Suggestions for Authors

The paper focuses on the analysis of the effect of L-arginine on lipopolysaccharide (LPS)-induced inflammatory response and oxidative stress in IPEC-2 cells. The paper is interesting and acceptable for publication in the journal. However, some minor changes should be addressed by the authors in order to improve the clarity and the understanding.

1.The Introduction should be more referenced. Some references are out of date and should be replaced by some more recent ones. In particular, the authors missed some recent works on L-Arginine functionalization as stabilizer and bioactive molecule.

Answer: Thanks for the reviewer’s valuable and helpful suggestion. According to the reviewer’s valuable comments, the Introduction has been more referenced, and we have added some recent references to replace those out of date and given more introduction of L-arginine.

2.The authors should report the acronyms the first time they appear in the text.

Answer: Thanks for your carefully and patiently reviewing our manuscript. We have carefully checked the whole manuscript throughout and reported the acronyms the first time they appear in the text.

3. Figure 6 is not cited in the text and should be more discussed.

Answer: Thank you for carefully and patiently reviewing our manuscript. According to the reviewer valuable comments, Figure 6 has been cited in the text and we have given more discussion.

Reviewer #2:

Comments and Suggestions for Authors

The paper describes a well established experimental effect of L-Arginine on LPS induced inflammatrory model of porcine intestinal cell line. The choice of cell line is proper, meaning it is modeling a primary cell culture and suitable for experiments of inflammatory models. The authors claim that in their experiments LPS had the usual inflammatory effects on cells, which was mainly protected by the co-administration of L-Arginine. They state that the protective effects of the above amino acid is partly carried out via an Arg-1 mediated signaling pathway.

The methods used in the paper are up-to-date, modern, well established and reasonable.

Comments and questions:

1.     1)How were the concentrations and durations of the treatments chosen? 2)How are these related to the in vivo circumstances? 3)Arginase is an enzyme of the urea cycle, expressed mainly in the liver. What is the role of arginase in the intestine? 4)The elevation of the level of arginase can be a cell signaling effect or merely an answer for the administration of its substrate?

Answer: 1) According to some related references, we chose 100ng/mL LPS to treated IPEC-J2 cells for 2, 6, 12, 24 and 48 h, we found that LPS decreased the cell viability at all of these treatment time, but there was no significant difference of cell viability among these treatments. Then 100, 250 and 1000ng/mL LPS were incubated with IPEC-J2 cells, and found that 100 and 250 ng/mL had significantly stimulated effects. Thus, we chose 100ng/mL LPS as the final treatment concentration. LPS-treated cells were co-incubated with L-arginine (100, 250,500 μM) for 2, 12, 24 and 48 h, after the treatment, we collected the cells to investigate the cell viability. At 2 h and 12 h, there was no difference of the numbers of IPEC-J2 cells in100, 250 and 500 μM L-arginine groups, as compared with those in LPS without L-arginine administration group. At 24 h, cell numbers were significantly increased with 250 and 500 μM L-arginine compared with LPS alone, and cell numbers in 500 μM L-arginine group were higher than those in 250 μM L-arginine. At 48 h, cell numbers were significantly increased with L-arginine concentrations from 100 to 500 μM compared with LPS alone, but there was no difference among the 100, 250 and 500 μM L-arginine groups. Thus, we thought that 24 h and 500 μM were optimal treatment time and concentration. 2) The L-arginine concentrations from 100 to 500 μM were physiological concentrations in plasma [1-2].

3) Previous study demonstrated that enterocytes of post-weaning pigs expressed arginases (including arginase-1 and arginase-2) to actively degrade L-arginine into ornithine and urea, with ornithine being further converted into proline by ornithine aminotransferase and pyrroline-5-carboxylate reductase [3-4]. 4) Arginase 1 is well known for its physiological role in hydrolyzing L-arginine to ornithine and urea, resulting in synthesis of polyamines, which are important regulators of cellular proliferation and tissue repair and anti-inflammatory responses [5-6]. In present study, arginase-1 expression was increased by L-arginine administration, while pretreatment of IPEC-J2 cells with the arginase 1 inhibitor abrogated anti-inflammatory effect of L-arginine. Thus, these data might indicate that L-arginine serves as a substrate for activating the activity of arginase-1, and exhibit its protective effect via arinase-1 (or polyamines: the downstream bioactive molecule). Thus, the elevation of the level of arginase-1 might also be a cell signaling effect.

[1] Kong X, Tan B, Yin Y, Gao H, Li X, Jaeger LA, Bazer FW, Wu G. L-Arginine stimulates the mTOR signaling pathway and protein synthesis in porcine trophectoderm cells. J Nutr Biochem. 2012.23,1178-83.

[2] Wu Z, Hou Y, Hu S, Bazer FW3, Meininger CJ, McNeal CJ, Wu G. Catabolism and safety of supplemental L-arginine in animals. Amino Acids. 2016.48,1541-52.

[3] Flynn NE, Meininger CJ, Kelly K, Ing NH, Morris SM Jr, Wu G.Glucocorticoids mediate the enhanced expression of intestinal type II arginase and argininosuccinate synthase in postweaning pigs. J Nutr.1999. 129,799-803.

[4] Wu G, Knabe DA, Flynn NE, Yan W, Flynn SP.Arginine degradation in developing porcine enterocytes. Am J Physiol Gastrointest Liver Physiol. 1996. 271,913-919.

[5] Morris SM Jr.Arginine metabolism: boundaries of our knowledge. J Nutr. 2007. 137,1602-1609.

[6] Taylor MD, Harris A, Nair MG, Maizels RM, Allen JE. F4/80+ alternatively activated macrophages control CD4+ T cell hyporesponsiveness at sites peripheral to filarial infection. J Immunol.2006. 176, 6918-6927.

2.How is the described pathway related to the NO production?

Answer: L-arginine serves as a common substrate of both nitric oxide (NO) synthase (NOS) and arginase [1]. Previous studies demonstrated that arginase and NOS competed for the L-arginine. Reduction or complete abolishment of arginase-1 activity is known to significantly enhance NOS-dependent NO production and increase inflammatory response [2-3]. Arginase-1 upregulation can attenuate NO production by iNOS in myeloid cells, thus reducing oxidative stress and inflammation [4]. In current study, our preliminary data suggested that the proliferation of IPEC-J2 cells regulated by LPS+L-arginine would not through NO signaling, since addition of NOS inhibitor, there was no significant change in cell cycle of IPEC-J2 (Supplemental Figure). Thus, we did not subsequently detect NO production.

[1] Wu G. Morris S M Jr. Arginine metabolism: nitric oxide and beyond. Biochem J.1998. 336, 1-17.

[2]Wijnands KA, Hoeksema MA, Meesters DM, van den Akker NM, Molin DG, Briedé JJ, Ghosh M, Köhler SE, van Zandvoort MA, de Winther MP, Buurman WA1, Lamers WH7, Poeze M. Arginase-1 deficiency regulates arginine concentrations and NOS2-mediated NO production during endotoxemia.PLoS One. 2014.9(1):e86135.

[3] Kim JH, Bugaj LJ, Oh YJ, Bivalacqua TJ, Ryoo S. Arginase inhibition restores NOS coupling and reverses endothelial dysfunction and vascular stiffness in old rats. J Appl Physiol.2009, 107: 1249-1257.

[4] Rath M, Muller I, Kropf P, Closs EI, Munder M. Metabolism via arginase or nitric oxide synthase: two competing arginine pathways in macrophages. Front. Immunol. 2014. 5, 532.

3.Some hard to understand misspellings are found in the text (e.g. in 2.3.: statics instead of statistics or rather statistical differences).

Answer: Thank you for pointing out the error. According to reviewer comment, the misspelling (statics) has been instead of statistics, and we have carefully proofread the whole manuscript throughout and corrected the error.

Once again, we would like to express our great appreciation to you and reviewers for comments on our paper.

Looking forward to hearing from you soon.

Best wishes,

Yueqin Qiu

Institute of Animal Science, Guangdong Academy of Agricultural Science, 510640 Guangzhou, Guangdong, China. E-mail: [email protected].

Phone/Fax: +86-20-85161287

Reviewer 2 Report

The paper describes a well established experimental effect of L-Arginine on LPS induced inflammatrory model of porcine intestinal cell line. The choice of cell line is proper, meaning it is modeling a primary cell culture and suitable for experiments of inflammatory models. The authors claim that in their experiments LPS had the usual inflammatory effects on cells, which was mainly protected by the co-administration of L-Arginine. They state that the protective effects of the above amino acid is partly carried out via an Arg-1 mediated signaling pathway.

The methods used in the paper are up-to-date, modern, well established and reasonable.

Comments and questions: How were the concentrations and durations of the treatments chosen? How are these related to the in vivo circumstances? Arginase is an enzyme of the urea cycle, expressed mainly in the liver. What is the role of arginase in the intestine? The elevation of the level of arginase can be a cell signaling effect or merely an answer for the administration of its substrate?

How is the described pathway related to the NO production?

Some hard to understand misspellings are found in the text (e.g. in 2.3.: statics instead of statistics or rather statistical differences).

Author Response

(The authors gave the same response as above.)
